# Novel Flavin Mononucleotide-Functionalized Cerium Fluoride Nanoparticles for Selective Enhanced X-Ray-Induced Photodynamic Therapy

**DOI:** 10.3390/jfb15120373

**Published:** 2024-12-10

**Authors:** Anastasia I. Kornienko, Maria A. Teplonogova, Marina P. Shevelyova, Matvei A. Popkov, Anton L. Popov, Vladimir E. Ivanov, Nelli R. Popova

**Affiliations:** 1Institute of Theoretical and Experimental Biophysics, Russian Academy of Sciences, Pushchino 142290, Russia; an.kor7@yandex.ru (A.I.K.); antonpopovleonid@gmail.com (A.L.P.); iwe88@rambler.ru (V.E.I.); 2Kurnakov Institute of General and Inorganic Chemistry, Russian Academy of Sciences, Moscow 119334, Russia; m.teplonogova@gmail.com (M.A.T.); ma_popkov@igic.ras.ru (M.A.P.); 3Pushchino Scientific Center for Biological Research of the Russian Academy of Sciences, Institute for Biological Instrumentation, Pushchino 142290, Russia; marina.shevelyova@gmail.com

**Keywords:** cerium fluoride nanoparticles, flavin mononucleotide, photosensitizer, X-ray-induced photodynamic therapy, cancer treatment

## Abstract

X-ray-induced photodynamic therapy (X-PDT) represents a promising new method of cancer treatment. A novel type of nanoscintillator based on cerium fluoride (CeF_3_) nanoparticles (NPs) modified with flavin mononucleotide (FMN) has been proposed. A method for synthesizing CeF_3_-FMN NPs has been developed, enabling the production of colloidal, spherical NPs with an approximate diameter of 100 nm, low polydispersity, and a high fluorescence quantum yield of 0.42. It has been demonstrated that CeF_3_-FMN NPs exhibit pH-dependent radiation-induced redox activity when exposed to X-rays. This activity results in the generation of reactive oxygen species, which is associated with the scintillation properties of cerium and the transfer of electrons to FMN. The synthesized NPs have been demonstrated to exhibit minimal cytotoxicity towards normal cells (NCTC L929 fibroblasts) but are more toxic to tumor cells (epidermoid carcinoma A431). Concurrently, the synthesized NPs (CeF_3_ and CeF_3_-FMN NPs) demonstrate a pronounced selective radiosensitizing effect on tumor cells at concentrations of 10^−7^ and 10^−3^ M, resulting in a significant reduction in their clonogenic activity, increasing radiosensitivity for cancer cells by 1.9 times following X-ray irradiation at a dose of 3 to 6 Gy. In the context of normal cells, these nanoparticles serve the function of antioxidants, maintaining a high level of clonogenic activity. Functional nanoscintillators on the basis of cerium fluoride can be used as part of the latest technologies for the treatment of tumors within the framework of X-PDT.

## 1. Introduction

Photodynamic therapy (PDT) is a relatively new and minimally invasive approach to cancer treatment [1,2,3,4]. The principle of this method is based on the selective accumulation of a photosensitizing agent (PS) in tumor tissue. The tissue is then exposed to light of a specific wavelength, resulting in the generation of reactive oxygen species (ROS), which have a cytotoxic effect on tumor cells [5,6,7]. However, PDT is not practical for treating deep-seated and large tumors because even in the near-infrared range, light can only travel less than 1 cm in tissue. This limitation has largely limited the use of PDT in clinical practice [5,7,8]. In 2006, it was proposed to use X-rays, with their high tissue penetration, to activate PS, and this approach was termed X-ray-induced PDT (X-PDT) [9,10]. The principle of X-PDT is based on the use of an energy transducer to convert X-rays into optical luminescence and initiate the radiotherapy and PDT processes [11,12,13,14].

Conventional PS used in PDT cannot be effectively activated by X-rays. Therefore, a physical transducer, scintillators, must be used to absorb the X-ray energy and transfer it to the photosensitizer to produce cytotoxic ROS, in particular singlet oxygen (^1^O_2_), which is necessary for tumor destruction. In this context, the development of new efficient and bioavailable scintillators is important. The current clinical use of established X-ray contrast agents includes iodinated molecules and barium sulfate solutions. However, these agents offer limited information, are not suitable for novel X-ray imaging techniques, and pose safety concerns [10]. Despite the recent interest in creating scintillators for X-PDT, only for RiMO-301 (radio-immuno-metal–organic framework) technology was a phase 1 clinical trial initiated in patients with advanced tumors (https://clinicaltrials.gov/ct2/show/NCT03444714, accessed on 15 February 2024).

For this purpose, the use of scintillation nanomaterials, such as lanthanide fluoride nanoparticles (NPs), is promising [9,15,16,17,18,19,20,21,22,23]. These materials have low toxicity for normal cells and can effectively absorb ionizing radiation and re-emit it in the form of photons [14,24,25,26,27,28,29,30]. In our study, we propose a new combined PS based on cerium fluoride (CeF_3_) NPs functionalized with flavin mononucleotide (FMN) for X-PDT purposes. CeF_3_ is a highly efficient scintillator due to the partially resolved transitions (d → f) of Ce^3+^ ions, which are more intense than the forbidden f → f transitions in other lanthanides. The fluoride matrix provides high quantum yield and photostability [31,32]. CeF_3_ NPs have been shown to be bioavailable and provide remarkable protection against oxidative stress and vesicular stomatitis virus in DPSc cells [33]. They also exhibit radioprotective effects on hMSC cells and are radiosensitizers for MCF-7 cancer cells [34]. The X-ray luminescence of CeF_3_ has a broad peak at approximately 325 nm that overlaps with the absorption spectrum of flavin mononucleotide (FMN) with peaks at 220, 265, 375, and 445 nm [32,35].

FMN is a derivative of riboflavin (vitamin B2), a co-factor in various enzymatic reactions of flavoproteins, and can therefore be considered an endogenous PS [36,37,38]. It is practically non-toxic in concentrations up to 10 g/kg [39]. It has been shown that FMN can be used in PDT to treat melanoma, with 85–90% regression in mice 50 days after the procedure [39]. Concurrently, FMN and riboflavin have been demonstrated to exhibit photocatalytic activity when exposed to UV radiation, resulting in the formation of ROS (singlet oxygen and superoxide radical with a quantum yield of 0.49 and 0.009, respectively) [40]. This phenomenon has been linked to the inhibition of tumor growth. Furthermore, riboflavin has been proposed for use as a free radical scavenger during electron beam and gamma sterilization of allografts [41].

Therefore, the aim of this study is to synthesize CeF_3_ nanoparticles modified with FMN for potential use in X-PDT, as well as to study their physicochemical properties and evaluate cyto- and genotoxicity on cultures of mouse fibroblast NTCT L929 and human epidermoid carcinoma A431, also after exposure to X-rays.

## 2. Materials and Methods

### 2.1. Synthesis of CeF_3_ and CeF_3_-FMN Nanoparticles

The synthesis of CeF_3_ NPs was conducted via precipitation in an alcoholic medium [42]. The synthesis of CeF_3_-FMN NPs was conducted using a modified protocol, employing a mixture of cerium (III) chloride heptahydrate (Alfa Aesar, Haverhill, MA, USA), isopropyl alcohol (Aldrich, W292907, St. Louis, MI, USA), and hydrofluoric acid (Component-reaktiv) as the starting material, with the subsequent addition of flavin mononucleotide (riboflavin-5′-phosphate, FMN) (Pharmstandard, Dolgoprudny, Russia). The resulting precipitate was filtered, washed with pure isopropyl alcohol, dried to a pasty state, and dispersed in 110 mL of water using an ultrasonic bath. The nanoparticle sol stock concentration is 45.2 × 10^−3^ M = 10.1 mg/mL for CeF_3_ and 42.55 × 10^−3^ M = 8.2 mg/mL CeF_3_-FMN.

### 2.2. Characterization of NPs

The hydrodynamic diameter and zeta-potential of the resulting NPs were measured using dynamic light scattering (DLS) and electrophoretic light scattering (ELS) with a BeNano 90 Zeta (BetterSize, Shanghai, China). The measurements were carried out at 25 °C. Each measurement represented an average of 15 runs (the number of runs was determined automatically by the instrument). The software package enables the estimation of diameters through the utilization of a distribution analysis model. The samples were measured on at least three occasions, with an average measurement error of approximately 5%. Scanning transmission electron microscopy (STEM) was performed using a Tescan Amber GMH (Brno, Czech Republic) scanning electron microscope equipped with an R-STEM detector at an accelerating voltage of 30 kV. The samples were dispersed in MQ water deposited on the formvar/carbon Cu grid (Ted Pella Inc., Redding, CA, USA) and dried at room temperature. The chemical composition analysis (energy dispersive X-ray analysis, EDX) of the samples was performed using a Carl Zeiss NVision 40 (Carl Zeiss, Jena, Germany) field emission scanning electron microscope equipped with an Oxford Instruments INCA (Oxford Instruments, Abingdon, UK) (80 mm^2^) detector at an accelerating voltage of 20 kV. X-ray phase diffraction (XRD) of the sols dried at 50 °C was performed using a Bruker (Billerica, MA, USA) D8 Advance powder X-ray diffractometer (CuKα radiation). The diffractograms were recorded at 0.02° s^−1^ for 2θ values ranging from 20 to 80° and accumulation for 60–90 min. Diffractograms were indexed using the International Center for Diffraction Data (ICDD) PDF2 database from 2012. The size of the coherent scattering regions (OCD) was calculated using the Scherrer equation, and peak profiles were approximated using Voigt pseudo-functions. UV–Vis absorption spectra of CeF_3_ and CeF_3_-FMN colloid solutions were recorded using standard quartz cells for liquid samples at 200–800 nm in a UV–Vis spectrophotometer (CARY 100 UV–Vis spectrophotometer (Varian, Inc., Palo Alto, CA, USA). Photoluminescence measurements CeF_3_ and CeF_3_-FMN colloid solutions were performed using a Cary Eclipse spectrofluorometer (Varian, Inc.) at room temperature (resolution: 0.5 nm; slit width: 2.5–10 nm).

### 2.3. Fluorescence Quantum Yield

The room temperature quantum yield of CeF_3_ solutions in ultrapure water (18 MΩ) was determined using the Cary Eclipse spectrofluorometer (Varian, Inc.) according to the manufacturer’s recommendations by the method described in [43,44]. Tyrosine (Dia-M, Moscow, Russia) solutions from (38 to 150)·10^−6^ M in ultrapure water were used as the quantum yield reference. The quartz cuvette with a pathlength of 1 cm was used. Fluorescence of tyrosine, CeF_3_, and CeF_3_-FMN NPs was excited at 265 nm, and emission spectra from (270 to 450) nm were recorded. Absorption at the excitation wavelength of the solutions was measured using a Cary 100 UV–Vis spectrophotometer (Varian, Inc.). Areas under fluorescence emission spectra for tyrosine, CeF_3_, and CeF_3_-FMN were obtained from fits of the spectra with log-normal curves [45] using LogNormal software version 1.0 (IBI RAS, Pushchino, Russia). Relative quantum yield was calculated as follows:(1)$$Q_{i}=Q_{Tyr}\cdot \frac{m_{i}}{m_{Tyr}}\cdot {\left(\frac{n_{i}}{n_{Tyr}}\right)}^{2},$$
where *Q_Tyr_* and *Q_i_* are the quantum yields; *m_Tyr_* and *m_i_* are the gradients of the plots of integrated fluorescence intensity against absorbance; *n_Tyr_* and *n_i_* are the refractive indices of the corresponded solvents; and subscripts *Tyr* and *i* denote the reference (tyrosine) and the sample (CeF_3_ and CeF_3_-FMN), respectively.

### 2.4. X-Ray Exposure

X-ray irradiation was conducted using the X-ray therapeutic machine RTM-13 (Mosrentgen, Moscow, Russia) at doses of 1, 3, and 5 for the NPs solution and 1, 2, 3, 4, 5, and 6 Gy for cell cultures, at a dose rate of 1 Gy/min, with a 200 kV voltage, 37.5 cm focal length, and a 20 mA current. The radiation doses were selected based on the data pertaining to median lethal doses (LD_50_) for cell cultures.

### 2.5. Acellular ROS Assay and Chemical Dose Enhancement Quantification

This method was previously described in greater detail in an earlier publication [45,46]. The 2′,7′-dichlorodihydrofluorescein diacetate (H_2_DCF-DA) powder (Sigma-Aldrich, St. Louis, MO, USA) was dissolved in 5 mM dimethyl sulfoxide (DMSO, PanEko Moscow, Russia) and stored at −20 °C for subsequent use. An initial solution of H_2_DCF was prepared by mixing 10 mM NaOH with four times the amount of H_2_DCF and incubating the mixture in the dark at room temperature for 30 min. Subsequently, a working solution of H_2_DCF (8 μM) was prepared in Tris-HCl buffer (pH 6.4, 7.2, or 8.0) and stored on ice. The nanoparticles (NPs) were diluted into Tris-HCl to achieve final concentrations of 2.5, 5, and 15 μM, respectively, and were added to 1.5 mL microtubes containing 0.1 mL Tris-HCl, 0.1 mL nanoparticle suspension, and 0.8 mL of the H_2_DCF working solution. Control samples were prepared by the addition of buffer in lieu of nanoparticles, and samples were irradiated with 1, 3, and 5 Gy of radiation. The dose range was determined on the basis of the observation that a single low dose of radiation, with a typical total dose of <5 Gy [25]. Following irradiation, the samples were subjected to centrifugation in their microtubes, after which the supernatant was transferred to transparent 96-well plates. Fluorescence was quantified using a microplate reader with excitation and emission wavelengths of 485 and 535 nm, respectively. The dose enhancement factors (DEF_ROS_) were calculated based on the fluorescence intensity (FI) as follows:(2)$$DEF_{ROS}=\frac{FI_{nGy\;with\;NP}\;-\;FI_{0Gy\;with\;NP}}{FI_{nGy\;without\;NP}\;-\;FI_{0Gy\;without\;NP}}.$$

### 2.6. Cell Culture

The experiments were conducted using a culture of mouse fibroblasts (NTCT L929) and human epidermoid carcinoma cells (A431), which were obtained from the cryostorage of the Theranostics and Nuclear Medicine Laboratory (ITEB RAS, Pushchino, Russia). The cells were cultivated in a culture medium comprising DMEM/F12 (1:1), supplemented with 50 μg/mL penicillin, 50 μg/mL streptomycin, 10% fetal bovine serum (FBS), and 1% l-glutamine. The cells were maintained at 37 °C in a humidified atmosphere containing 95% air and 5% CO_2_ in order to ensure optimal conditions for cell growth. To evaluate the cytotoxic and genotoxic effects of CeF_3_ and CeF_3_-FMN NPs, cells were co-incubated with the NPs at varying concentrations (10^−7^, 10^−6^, 10^−5^, 10^−4^, 10^−3^ M). The control groups were not treated with NPs. The cells in the control groups were not exposed to any of the samples.

### 2.7. MTT Assay

The viability of cells was evaluated through the use of MTT analysis, which assesses the capacity of cells to convert MTT salt (3-(4,5-dimethylthiazole-2-yl)-2,5-diphenyltetrazolium bromide) into formazan (absorption at 540 nm) through the action of cellular NAD(P)H-dependent oxidoreductases. The cells were seeded into 96-well plates at a density of 25,000–35,000 cells per cm^2^. Following a 24-, 48-, and 72-h co-culture of the cells in a culture medium containing nanoparticles, the medium was replaced with a solution of MTT (0.5 mg/mL) in DMEM/F12. Following a three-hour incubation period, the medium was replaced with dimethyl sulfoxide (DMSO) (PanEco, Moscow, Russia) and agitated for 10 min to facilitate the dissolution of formazan. Subsequently, the absorption of the solutions was quantified utilizing a Multiscan FC plate spectrophotometer (Thermo Fisher Scientific, Waltham, MA, USA). The obtained absorption values were calculated as a percentage of the control group values, and any deviations in the samples were presented as a standard deviation (SD).

### 2.8. Live/Dead Assay

The cytotoxic effect of NPs (CeF_3_, CeF_3_-FMN) was evaluated through the utilization of a live/dead assay. The cells were stained with a combination of fluorescent dyes that bind to either all cells’ DNA (Hoechst 33342, excitation 350 nm, emission 460 nm) or only dead cells (propidium iodide, excitation 535 nm, emission 615 nm). To this end, following the cultivation of the cells with nanoparticles for 24, 48, and 72 h, the culture media was replaced with a dye solution in DMEM/F12 and incubated for 15 min. Subsequently, the cells were washed with Hanks’ balanced saline solution (HBSS) and imaged using a ZOE imager (Bio-Rad, Hercules, CA, USA). The number of cells in three fields of view on three separate micrographs was counted using the ImageJ software 1.54k. Subsequently, the ratio of the number of dead cells to the total number of cells was calculated.

### 2.9. In Vitro Micronucleus Assay

The micronuclear assay was conducted by staining the cell nuclei with the fluorescent dye 4,6-diamidino-2-phenylindole (DAPI) at a concentration of 0.6 μg/mL (ServiceBio, Wuhan, China). The cells were co-incubated with NPs at concentrations ranging from 10^−6^ to 10^−3^ M for a period of 24 h. Following this, the cells were fixed and stained with DAPI, and the nuclei were analyzed quantitatively using a photofixation technique with a 63× oil lens on an inverted microscope (Zeiss 200 Axiovert, Oberkochen, Germany). The image processing was conducted using the GNU Image Manipulation Program (GIMP), and only 300 nuclei were randomly selected from each field for evaluation of micronucleus presence. The assessment was conducted in triplicate.

### 2.10. Clonogenic Assay

Cell cultures were co-incubated with 10^−7^ and 10^−3^ M nanoparticles for a period of 24 h. Following this, the medium was replaced and the cells were subjected to irradiation. Subsequently, cells at a concentration of 1500/well were seeded into 6-well plates (SPL LifeScience, Pocheon-si, Republic of Korea) in a DMEM/F12 culture medium containing 10% FBS and cultured under normal conditions for 7–8 days until colonies formed in the control group. The cells were washed with a balanced Hank’s saline solution, fixed in 4% paraformaldehyde solution (PFA) (PanReac AppliChem, Barcelona, Spain), and stained with 0.1% crystal violet (PanEko, Moscow, Russia). The number of colonies was quantified using the ImageJ software (National Institutes of Health, Bethesda, MD, USA).

In order to ascertain the survival rate, the number of colonies formed was divided by the number of seeded cells and subsequently normalized in relation to the seeding efficiency of the non-irradiated control sample [47]. A standard linear-quadratic formula was employed for the analysis of the data pertaining to clonogenic survival.


SF = e – (αD + βD^2^),
(3)

where SF is the fraction of surviving cells; D represents the radiation dose (Gy); and α and β are linear and quadratic empirical coefficients, respectively, that describe the radiosensitivity of cells.

### 2.11. Statistical Analysis

The experiments were conducted in 3–5 repetitions, with three independent repetitions for each NPs concentration. Experimental results were compared with intact control. Statistical analysis was performed using the methods of variation statistics (ANOVA, Student’s *t*-test). The mean values and the standard deviation (SD) of the mean were determined. The results are presented as a statistical mean value ± standard deviation. The significance of the deviations between the samples and the control was confirmed using Student’s *t*-test and the Mann–Whitney U test at 0.01 < *p* < 0.05 (*), 0.001 < *p* < 0.01 (**), and *p* < 0.001 (***). The obtained data were processed using the GraphPad 8.0 software.

## 3. Results

### 3.1. Synthesis, Properties, and Characterization of CeF_3_-FMN NPs

The synthesis procedures resulted in stable aqueous sols of CeF_3_ and CeF_3_-FMN NPs. The schematic representation of CeF_3_-FMN nanoparticles is shown in Figure 1. Scanning transmission electron microscopy (STEM) data indicate that the obtained CeF_3_ (Appendix A) and CeF_3_-FMN (Figure 2a) samples consist of nearly monodisperse NPs with an average size of 15–30 nm and were predominantly spherical in shape. The ζ-potential of CeF_3_ and CeF_3_-FMN NPs sols was +39 mV and +44 mV, respectively, confirming their high stability (Appendix A). According to DLS data, the mean hydrodynamic diameter is 66 nm for CeF_3_ and 74 nm for CeF_3_-FMN NPs (Appendix A). The difference between the mean particle size values measured by STEM and DLS indicates some degree of particle agglomeration in colloidal solutions. Local energy-dispersive X-ray spectroscopy data for CeF_3_ and CeF_3_-FMN samples confirm the presence of Ce and F (Appendix A). According to the results of the X-ray diffraction analysis (Figure 2c), the obtained diffraction patterns correspond to those of cerium fluoride (CeF_3_, spatial group P63/mcm, PDF2 card No. 8–45). In the case of particles modified with FMN, there is a slight shift in the reflections towards large angles of 2θ. According to Bragg’s law, a shift towards large angles indicates a decrease in the unit cell parameters. The size of the coherent scattering region calculated by the Scherrer formula was 63 ± 12 nm and 18.4 ± 0.7 nm for the CeF_3_ and CeF_3_-FMN samples, respectively.

The main UV absorption peak of CeF_3_ NPs is located at 250 nm (Figure 2d). According to the literature data, the UV absorption spectra of Ce^3+^ ions have a maximum at 253.6 nm with a molar extinction coefficient of 685 M^−1^ cm^−1^ [48]. The UV spectra of CeF_3_-FMN NPs confirm that the change in the absorption spectrum between 350 and 500 nm is observed compared to the unmodified samples. This may be due to the overlap of the absorption spectra for CeF_3_ and FMN (absorption peaks at 445 nm, 375 nm, 265 nm, and 225 nm) (Figure 2d). The CeF_3_-FMN sol shows (Figure 2e) an intense luminescence in the green region (having broad emission bands with maxima at 523–527 nm) of the visible spectrum under UV irradiation (ex 250–270 nm). This is in agreement with literature data on FMN emission spectra [49]. The same band is observed in the excitation spectrum of CeF_3_ NPs emitting only in the UV region (em 290–500 nm). Figure 2e shows a possible overlap between the absorption spectrum of FMN and the emission spectrum of CeF_3_. This indirectly supports the possibility of energy transfer from CeF_3_ to FMN within the NPs.

At the next stage, data on the quantum yield of NPs was obtained. Fluorescence emission spectra of tyrosine solutions as well as CeF_3_ and CeF_3_-FMN NPs sols and linear plots of integrated fluorescence intensity against absorbance are presented in Figure 3. The data pertaining to the absorbance and integrated fluorescence intensity values are presented in Appendix A. Quantum yield for tyrosine was taken to be *Q*_Tyr_ = 0.13 ± 0.01 [50]; the refractive indices ratio ${\left(\frac{n_{i}}{n_{Tyr}}\right)}^{2}$ was assumed to be 1. According to Equation (1), the relative quantum yields for CeF_3_ and CeF_3_-FMN are *Q*_CeF3_ = 0.42 ± 0.02 and *Q*_CeF3-FMN_ = 0.11 ± 0.02, respectively.

### 3.2. Reactive Oxygen Species Generation in Nanoparticle Suspensions

The data presented in Figure 4 demonstrates that CeF_3_ and CeF_3_-FMN NPs (0.25 × 10^−3^, 0.50 × 10^−3^, and 1.50 × 10^−3^ M) are capable of generating ROS in water solutions (pH 6.4, 7.2, and 8.0) upon exposure to X-rays. To ascertain the dose-dependent formation of ROS under the influence of X-ray radiation and NPs, irradiations were conducted at doses of 1, 3, and 5 Gray. Figure 4 illustrates that in acidic solutions (pH 6.4), the addition of CeF_3_ NPs (1.50 × 10^−3^ M) results in a 14-fold increase in ROS levels relative to the control, while CeF_3_-FMN NPs exhibit a 14.7-fold increase. In a neutral pH of 7.2, the DEF_ROS_ index demonstrates an increase of 8 and 7 times for CeF_3_ and CeF_3_-FMN, respectively, in comparison to the control. In an alkaline solution (pH 8.0), the NPs display only weak antioxidant properties, resulting in a decrease in the ROS level.

### 3.3. Cyto- and Genotoxicity Study of CeF_3_ and CeF_3_-FMN NPs

The biocompatibility of CeF_3_ and CeF_3_-FMN NPs was evaluated through the utilization of the MTT assay and live/dead assay, employing normal (mouse fibroblasts NCTC L929) and cancer (human epidermoid carcinoma A431) cell cultures. The MTT assay for cellular metabolic activity is a standard method for assessing cell toxicity. It measures the enzymatic activity of intracellular mitochondrial NADPH-dependent oxidoreductases and provides insight into cell viability following interaction with the test substance [51]. The results of the MTT assay on normal NCTC L929 cells co-incubated with various concentrations of NPs for 24, 48, and 72 h are presented in Figure 5a. It was observed that the co-incubation of fibroblasts for 24 and 72 h with all of the studied NPs did not result in a notable increase in cell death in comparison to the control (Figure 5a). However, following a 48-h incubation period, a reduction in cell viability was observed, with an IC_20_ concentration of CeF_3_ (10^−5^ M) and CeF_3_-FMN NPs (10^−6^ to 10^−5^ M) being identified. The live/dead assay is a rapid and straightforward two-color assay that enables the determination of cell viability within a population based on plasma membrane integrity and esterase activity. The results demonstrated that the co-incubation of fibroblasts with all studied NPs for 24 to 72 h did not significantly increase cell death compared to the control (Figure 5a, right graph). No IC_20_ was identified within the concentration range under investigation.

According to the results of the study of the metabolic activity of A431 cancer cell cultures, we observe interesting results (Figure 5b). It has been demonstrated that the co-incubation of nanoparticles (NPs) for a period of 24 h does not result in a reduction in cell viability. Conversely, following a 48-h and 72-h incubation period, a notable decline in cell viability was observed upon co-incubation with NPs. The results of the live/dead assay indicate that co-incubation of A431 with NPs does not result in a notable increase in the proportion of dead cells (Figure 5b, right graph). No IC_20_ was identified within the concentration range of the nanoparticles. Nevertheless, in comparison to normal cells, a more pronounced toxic effect was observed in cancer cells.

Subsequently, the genotoxicity of CeF_3_ and CeF_3_-FMN NPs was assessed on normal (NCTC L929) (Figure 6a) and cancer (A431) (Figure 6b) cells using a micronucleus test. The method for assessing genotoxicity entails the detection of micronuclei, which indicate chromosomal damage, within the cytoplasm of interphase cells. The co-incubation of all the studied NP samples revealed no disruption of the cell nucleus morphology and no significant increase in micronuclei formation or nuclear fragmentation compared to the control group.

### 3.4. Effect of CeF_3_ and CeF_3_ + FMN NPs on Cell Survival After X-Ray Exposure

A comparative assessment of the impact of CeF_3_ and CeF_3_-FMN NPs on the survival and capacity to form colonies of normal NCTC L929 and cancer A431 cells was conducted following X-ray exposure (Figure 7). A clonogenic assay, also known as a colony formation assay, is an in vitro cell survival assay based on the ability of a single cell to grow into a colony, that is, to undergo continuous proliferation [52]. This experimental approach is widely used to test the effects of NPs on cell growth and proliferative characteristics. It should be noted that the experiments were conducted using doses of 1, 2, 3, 4, 5, and 6 Gy. The radiation doses were selected based on the findings of a series of preliminary experiments, which aimed to determine the median lethal doses (LD_50_) for each cell culture at which the number of cells would decrease by at least 50% compared to the control group (unexposed).

Conventionally, the outcomes of colony formation assays are presented as so-called survival curves representing the survival fraction (SF), i.e., the number of colonies that are formed after treatment, as a function of radiation dose (D) [52]. The figure shows the dose–response survival curves (Figure 7). The α and β parameters, as well as their ratio in the linear quadratic formula (LQ model), are presented in Appendix A, depending on the concentration of NPs. To compare the radiosensitivity of cells and its changes after incubation with NPs, a D_0_ dose was calculated at which the clonogenic potential preserves 37% of the original number of cells [52] (Appendix A). Thus, D_0_ for NCTC L929 cells was 2.7 Gy for the control group; 4.5 and 4.8 Gy for CeF_3_ (10^−7^ and 10^−3^ M concentrations, respectively); and 6.9 and 5.3 Gy for CeF_3_-FMN NPs (10^−7^ and 10^−3^ M concentrations, respectively). For A431 cancer cells, D_0_ was 3.5 Gy for the control group; 2.5 and 1.8 Gy for CeF_3_ NPs (10^−7^ and 10^−3^ M concentrations, respectively); and 2.5 and 2.2 Gy for CeF_3_-FMN NPs (10^−7^ and 10^−3^ M concentrations, respectively).

## 4. Discussion

In our recent study, CeF_3_-modified FMN NPs were synthesized. As a result, an aqueous high-stability sol of CeF_3_-FMN NPs is formed (ζ-potential +44 mV, Appendix A). In order to confirm the size of these particles, STEM was performed. STEM micrographs suggested that the particles had a uniform size distribution and were sub 15–30 nm in spherical particle form (Figure 2a,b and Appendix A, Appendix A). However, the behavior of particles may vary in colloidal form, as particles may aggregate in a kinetically driven process through the formation of clusters. The hydrodynamic diameter and size distribution of the particles were measured via DLS (Appendix A). The chemical composition of NPs has been confirmed through a variety of methods (EDX Appendix A, XRD Figure 2c). We have demonstrated that the obtained NPs meet the criteria for the production of nanomaterials for biomedical applications. Aggressive media and precursors were not used in the synthesis of NPs. We have shown that NPs have a round shape and are smaller than 100 nm in size, making them potentially bioavailable [53]. Most in vitro studies have shown that the maximum cellular uptake is within the 10–60 nm range, regardless of core composition or surface charge [54]. After modification of CeF_3_ NPs (ex 250 nm) with FMN, we observe that UV spectra of CeF_3_-FMN NPs confirm the change in the absorption spectrum between 350 and 500 nm. This may be due to the overlap of the absorption spectra for CeF_3_ and FMN (ex 445 nm, 375 nm, 265 nm, and 225 nm) (Figure 2).

CeF_3_ is regarded as one of the most effective scintillators for use in biomedical applications [15]. Scintillators are compounds that are capable of emitting photons when they absorb ionizing radiation of various types, including X-rays. CeF_3_ NPs function as an effective scintillator, whereby upon X-ray irradiation, UV light is emitted due to fluorescence at a wavelength of 325 nm. The scintillation properties of cerium-doped fluoride compounds are described in detail in the literature [55,56]. The prominent luminescence properties of Ce^3+^ species (in contrast to those of Ce^4+^) are attributed to the 4f–5d transition of Ce^3+^ [42]. Furthermore, additional absorption and excitation bands at higher energies have been assigned to Ce 4f→6s and F 2p→Ce 5d transitions [42,57]. In the context of fast scintillators, Ce^3+^-doped or -based materials are of particular interest due to the fast and generally intense parity-allowed 5d→4f transition, which results in a blue UV fluorescence contingent on the host material. The fast scintillation decay occurs within a range of 20 ns, with a light yield between 5000 and 8000 photons/MeV. Additionally, photons with energies of 4–5 eV can only excite intracenter 4f–5d transitions of cerium ions [56]. The CeF_3_ exhibits luminescence under UV light, displaying a faint purple and an intense green region when FMN is added (Figure 2). Figure 2 illustrates a potential overlap between the absorption spectrum of FMN and the emission spectrum of CeF_3_. This provides indirect evidence to support the hypothesis that energy transfer from CeF_3_ by scintillation to FMN within NPs is a possible mechanism. It is postulated that following exposure to X-ray radiation, FMN will become photo-activated and will fluoresce as a consequence of scintillation radiation emitted by cerium fluoride [32,58].

The quantum yield of fluorescence is an important parameter in the evaluation of scintillating materials, the values of which range from 0 to 1 (Figure 3). The higher the value of this parameter, the more absorbed energy is transformed into radiation. We have shown that relative quantum yields for CeF_3_ in NPs are *Q*_CeF3_ = 0.42 ± 0.02 and *Q*_CeF3-FMN_ = 0.11 ± 0.02, respectively. This allows us to conclude that in the nanosystem we synthesized, FMN efficiently absorbs the light emitted by excited CeF_3_. Also, these results are in line with the literature data, which suggests that CeF_3_ is a fantastic luminescent material [55] with high photoelectric conversion efficiency and long excited state lifetime [59]. The quantum yield of CeF_3_ NPs- is 0.31 [32] and 0.15 [60]; the quantum yield for Ce_0_._1_La_0_._9_F_3_/LaF_3_ is 0.17, and for CeF_3_/LaF_3_ it is 0.21 [61]. Thus, in our study, we successfully synthesized a compound with a high quantum yield.

The analysis of the cytotoxicity revealed that when comparing the effects of CeF_3_ and CeF_3_-FMN NPs on normal NCTC L929 cells and transformed A431 cells (Figure 5), it can be concluded that the studied NPs do not exhibit a significant toxic effect on these cell lines within the concentration range studied (10^−6^ to 10^−3^ M). According to the MTT assay results, after 48 h of co-incubation of NCTC L929 cells with NPs, a slight decrease in viability was observed, which was recovered after 72 h. However, when NPs were incubated with cancer cells, a more significant cytotoxic effect was noted, leading to a substantial decrease in viability down to IC_20_ or lower. No significant cell death (by live/dead assay) was detected in either cell type; IC_20_ was not observed in this concentration range of NPs. We also found that all the NPs under study do not have a genotoxic effect on either type of cell, according to the results of the in vivo micronucleus assay (Figure 6). In study [33], it was shown that CeF_3_ NPs do not exhibit cytotoxicity and genotoxicity to DPSc, even at high concentrations (10^−4^ M). The toxicity of NPs can be attributed to various factors, including the size, shape, and crystallinity of the particles, as well as the surface chemistry and chemical composition of the materials used in their synthesis. Previously, it has been suggested that CeF_3_ may be non-toxic because the toxicity of fluorine-containing inorganic substances typically depends on their solubility in water. CeO_2_ NPs and CeF_3_ NPs have a low solubility in both water and biological fluids [30].

However, it was observed that CeF_3_ and CeF_3_-FMN NPs demonstrated a more pronounced toxicity towards A431 cells, resulting in a higher rate of cell death compared to normal cells (Figure 5). As has been demonstrated previously, CeF_3_ NPs have the dual effect of stimulating the proliferative activity of normal cells (hMSC) and suppressing that of cancer cells (MCF-7) [34]. Interestingly, the water-soluble riboflavin molecule is transported into the cell via the riboflavin carrier protein (RCP), which is located in the cell membrane. Previous studies have demonstrated that riboflavin uptake and trafficking are significantly higher in A431 cells [62] than in healthy cells [63]. Also, it is established that the metabolic rate, proliferative activity, and levels of ROS in cancer cells are significantly elevated in comparison to normal cells [64]. The observed effect on cancer cells may be attributed to the prevailing acidic pH in the cancer cell microenvironment (pH 5.6–7.0). The pH of a normal tissue interstitium is typically within the range of 7.2 to 7.5. It is acknowledged that pH is a pivotal factor in the therapeutic efficacy of cerium-containing nanoparticles [65]. In acidic conditions, an excess of H^+^ can inhibit the conversion of Ce^4+^ to Ce^3+^, which catalyzes the decomposition of the absorbed surface of H_2_O_2_. This, in turn, disrupts the repeated action of active catalytic centers and blocks antioxidant cycles. Consequently, elevated levels of H_2_O_2_ are accumulated within cancer cells, which ultimately results in their demise. The study also demonstrated that CeF_3_ and CeF_3_-FMN NPs are capable of generating ROS in aqueous solutions following exposure to X-ray radiation (1, 3, 5 Gy) at varying pH levels (pH 5.4–8.0) (Figure 4). It has been demonstrated (Figure 4) that the oxidant properties of nanoparticles exhibit a positive correlation with decreasing pH values. This finding is in accordance with the data obtained from other studies [66,67]. CeF_3_ NPs significantly reduce the content of hydrogen peroxide and hydroxyl radicals after X-ray exposure. The catalytic performance of NPs is contingent upon the non-stoichiometric ratio of Ce^3+^/Ce^4+^ on their surface. In particular, cerium is capable of readily adjusting its electronic configuration in response to the immediate microenvironment. It is generally accepted that cerium-containing NPs with a predominant content of cerium in the trivalent state on the surface provide their SOD mimetic activity under neutral and acidic conditions. This can be observed in their ability to catalyze the dismutation of superoxide radicals (O_2_^−^) into H_2_O_2_ [68]. Cerium oxide NPs exhibit high activity in the degradation of H_2_O_2_ in a neutral pH medium. However, their effectiveness is reduced under acidic conditions [27,62]. It is important to acknowledge that a number of studies have demonstrated that different cell types exhibit varying degrees of internalization of NPs [69,70,71,72,73,74,75,76]. For example, overexpression of epidermal growth factor in A431 cells has been demonstrated to facilitate the cellular uptake of polystyrene NPs (45 nm in diameter) via clathrin-mediated endocytosis [77]. Our preliminary research has indicated that cerium oxide nanoparticles (3–30 nm) accumulate in CTC L929 cells to a greater extent than in MNNG/HOS cancer cells [63,78]. Further research is required to elucidate the patterns of internalization of CeF_3_-FMN NPs into different cell types.

Figure 7 and Appendix A show that NPs, as a function of NP concentration, reduce the radiosensitivity of normal cells and increase it in cancer cells after exposure to X-rays. Moreover, a more pronounced effect is observed for CeF_3_-FMN NPs, reducing the radiosensitivity of normal cells by almost 2.5 times and increasing radiosensitivity for cancer cells by 1.9 times. NPs interfere with the cellular repair processes, and evaluating the change in the α/β parameter is useful for quantifying the radiobiological impact of specific NPs [79]. A change in the α/β ratio could indicate any radiosensitivity changes in the system caused by the NPs. Also Figure 7 shows that the dose–response curve has a different appearance. For normal cells, the curve is more curved, whereas for cancer cells it is straighter. A ‘curvy’ dose–response curve will obviously have a small value of α/β, whilst a rather straight curve will have a large value of α/β [80]. The α and β parameters describe the radiosensitivity of cells. The linear parameter α is related to the initial region of the LQ model curve and represents sensitivity to low doses. As the dose increases, the quadratic term β becomes more significant, increasing the curvature. The linear (α) and quadratic (β) parameters from the LQ model could be used to characterize the NP enhancement effect. Based on the above, we can conclude about the different radiosensitivity of NCTC L929 and A431 cells. Nanoparticles affect the radiosensitivity of cells, making normal cells less responsive to radiation when doses are increased (the curve becomes straighter) and, on the contrary, making cancer cells more sensitive (the curve becomes more ‘curvy’).

X-ray irradiation results in the radiolysis of water, which in turn gives rise to the formation of a multitude of different types of ROS. This results in an increased demand for the antioxidant system of the cells. In our case, this effect is likely due to the increased formation of ROS, in particular singlet oxygen, under the influence of CeF_3_-FMN NPs and X-ray radiation [81,82,83,84,85]. This may explain why cells treated with the NPs have a higher value of β. We observe this effect in our results (Figure 4), where there is an increase in the level of ROS formation directly proportional to an increase in the concentration of CeF_3_-FMN NPs under the action of radiation. Moreover, the NPs may interact directly with the incoming ionizing radiation or any of the intermediate chemical products to alter the final spectrum and yield of ROS, accounting for the localized effect of a cascade of Auger electrons [86]. Cerium-containing NPs mimic enzymes associated with ROS that protect normal cells from oxidative stress and induce reactive ROS in the slightly acidic tumor microenvironment to trigger cancer cell death [87] after exposure to X-rays [34]. Nanoceria generates molecular oxygen, which relieves tumor hypoxia [88], leading to tumor cells becoming more sensitive to photodynamic and radiation therapy. Nevertheless, this mechanism, or radiation-induced sensitization, cannot be the sole explanation for the selective therapeutic effect on tumor cells, given the relative importance of other factors, in particular Ce dissolution and the role of anions in the microenvironment, remains insufficiently studied.

## 5. Conclusions

The combination of PDT and RT methods in X-PDT has demonstrated considerable potential for the treatment of tumors. In this paper, we put forward a novel nanoscintillator-photosensitizer combination based on CeF_3_ NPs modified with FMN, which we propose for use within the X-PDT framework. A comprehensive analysis of the physicochemical properties of the NPs was conducted using STEM (scanning transmission electron microscopy), EDX (energy dispersive X-ray spectroscopy), and XRD (X-ray diffraction) analysis, and it was demonstrated that the NPs met the requisite specifications. The spectral characteristics of the synthesized nanoparticles have indirectly confirmed the possibility of excitation energy transfer from CeF_3_ to FMN. It has been demonstrated that the quantum yield of fluorescence of the low-frequency CeF_3_ is 0.42, which characterizes it as an efficient scintillator.

The objective of this study was to investigate the influence of nanoparticles on the level of ROS, cyto- and genotoxicity, and the proliferative activity of normal and cancer cells under exposure to X-rays, in comparison to CeF_3_ NPs. It has been demonstrated that CeF_3_ and CeF_3_-FMN, when subjected to X-ray radiation in an acidic environment, display pro-oxidant properties when present in buffer systems. These data are corroborated by the data obtained from the clonogenic analysis. It is hypothesized that the NPs in the microenvironment of tumor cells can generate a greater number of ROS, which is a crucial factor in the efficacy of the sensitizer during PDT. The effects of CeF_3_ and CeF_3_-FMN NPS on normal (NCTC L929) and transformed (A431) cells were assessed. It was found that they did not show geno- or cytotoxicity in the concentration range of 10^−6^ to 10^−3^ M for 24 h to 72 h. Furthermore, in view of the promising in vitro results, additional in vivo studies are imperative to corroborate the therapeutic effectiveness of the NPs in complex biological systems. This encompasses the issues of nanoparticle biodistribution, off-target effects, immune response, and long-term safety. These supplementary studies will facilitate a more profound comprehension of the underlying mechanisms and the broader applicability of CeF_3_-FMN NPs in X-PDT cancer therapy.

## Figures and Tables

**Figure 1 jfb-15-00373-f001:**
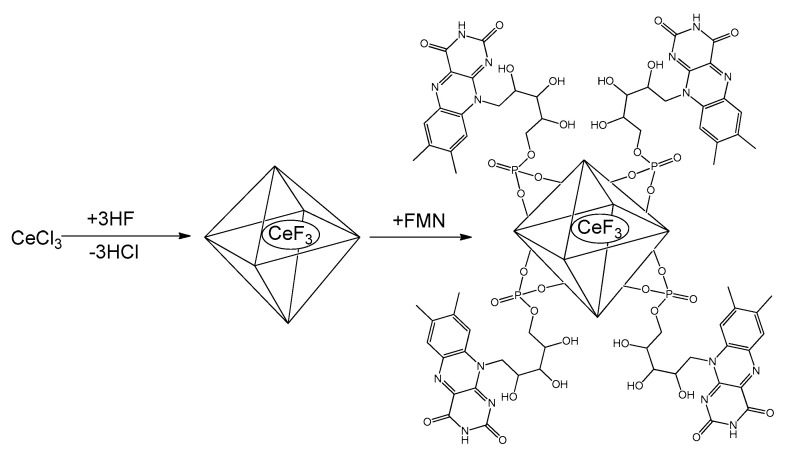
Schematic illustration of CeF_3_-FMN nanoparticles.

**Figure 2 jfb-15-00373-f002:**
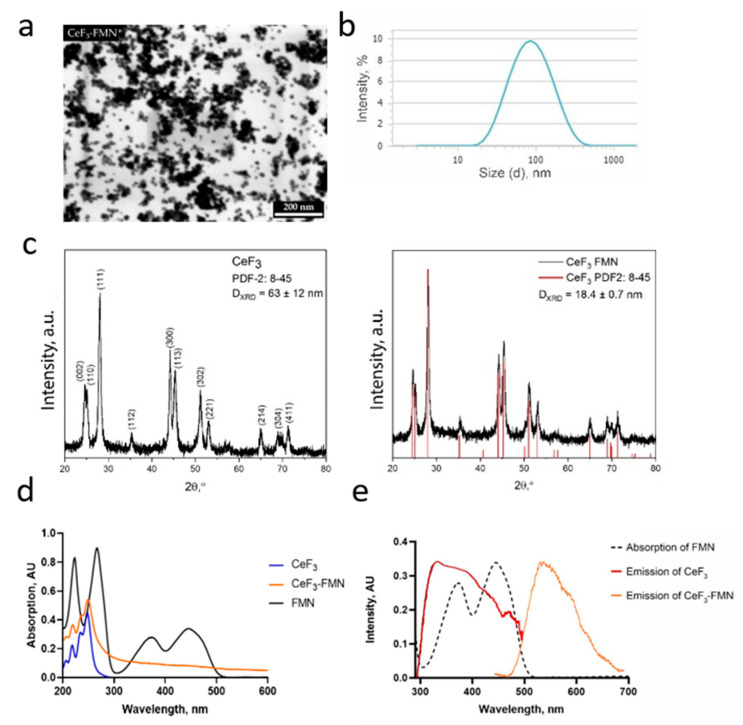
STEM image of CeF_3_-FMN NPs (**a**), hydrodynamic radii distribution (**b**), XRD patterns of NPs (**c**), absorption spectra of NPs and FMN solution (**d**), FMN absorption spectrum, and emission spectra of CeF_3_ and CeF_3_-FMN NPs sols (λex = 250–270 nm) (**e**).

**Figure 3 jfb-15-00373-f003:**
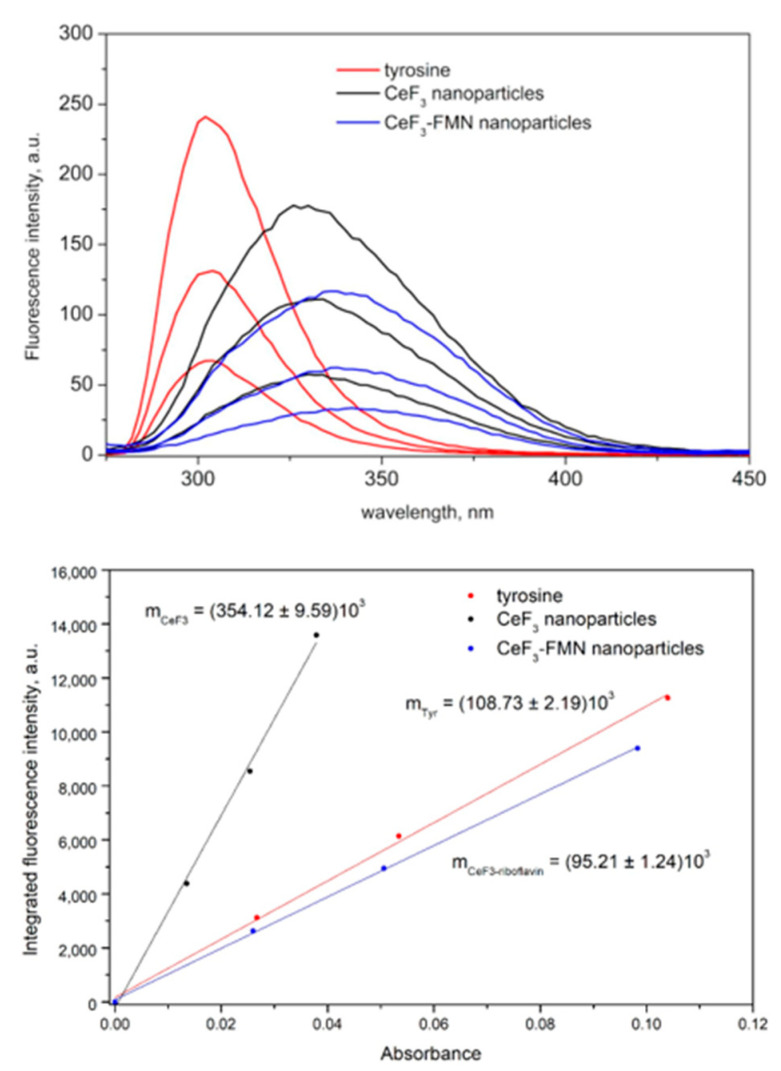
Fluorescence spectra (**top**) and integrated fluorescence intensity against absorbance (**bottom**) of tyrosine (red), CeF_3_ (black), and CeF_3_-FMN (blue) in water at 25 °C.

**Figure 4 jfb-15-00373-f004:**
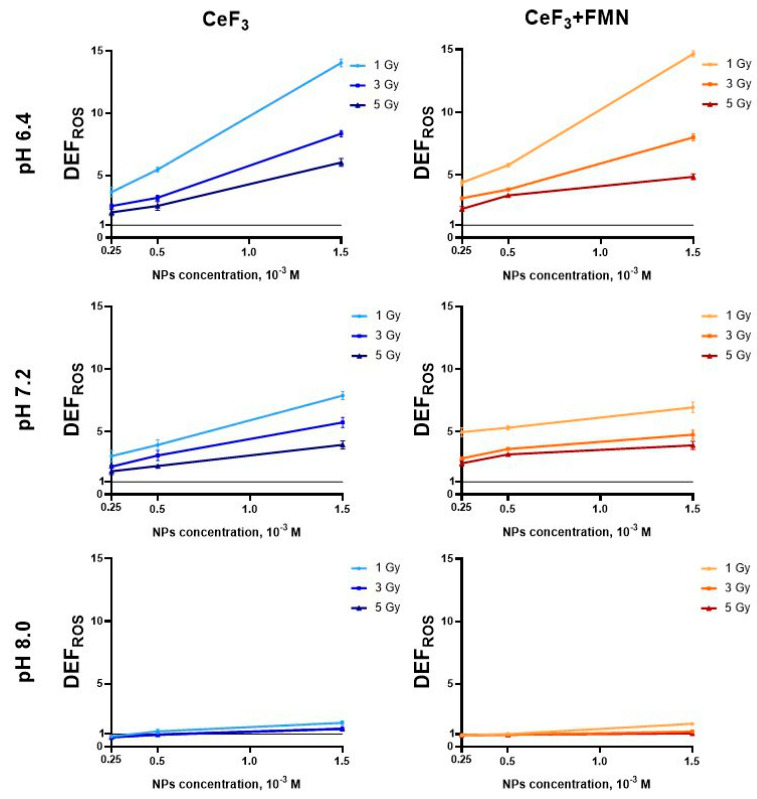
Reactive oxygen species generation in nanoparticle suspensions after X-ray exposure (1, 3, 5 Gy). Chemical dose enhancement factors (DEF_ROS_) for CeF_3_ and CeF_3_ + FMN as assessed using the acellular H_2_DCF-DA assay at pH 6.4, 7.2, and 8.0.

**Figure 5 jfb-15-00373-f005:**
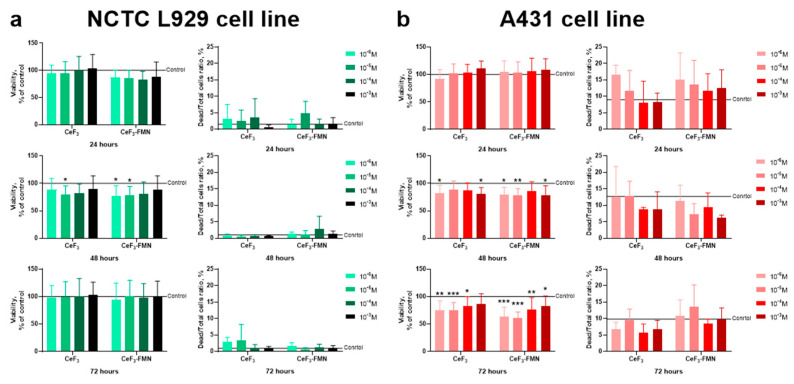
Cell viability and survival rate of mouse fibroblasts NCTC L929 (**a**) and human epidermoid carcinoma cells A431 (**b**) after 24, 48, and 72 h of co-incubation with CeF_3_ and CeF_3_-FMN NPs at concentrations of 10^−3^–10^−6^ M determined by the MTT assay and live/dead assay. Data are shown as M ± SD and were analyzed using the Mann–Whitney U test (*n* = 3); 0.01 < *p* < 0.05 (*), 0.0001 < *p* < 0.01 (**), and *p* < 0.001 (***).

**Figure 6 jfb-15-00373-f006:**
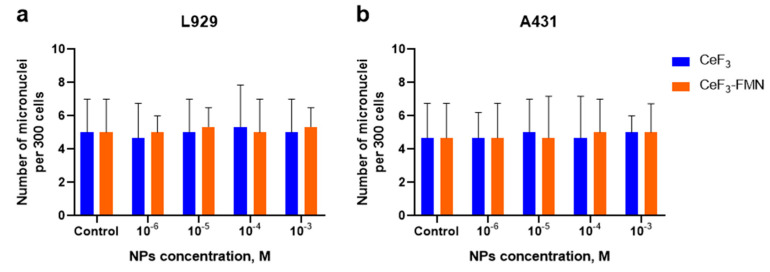
Micronucleus frequency of mouse fibroblasts L929 (**a**) and human epidermoid carcinoma cells A431 (**b**) after co-incubation with CeF_3_ and CeF_3_-FMN NPs at a concentration of 10^−6^–10^−3^ M. Data are shown as M ± SD and were analyzed using the Mann–Whitney U test (*n* = 3).

**Figure 7 jfb-15-00373-f007:**
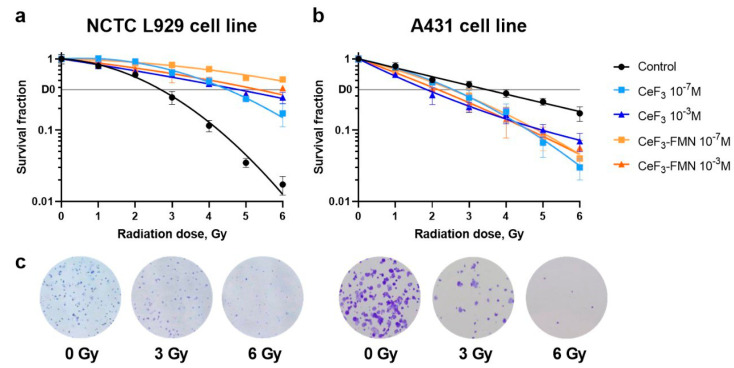
Dose–response curves of mouse fibroblasts NCTC L929 (**a**) and human epidermoid carcinoma cells A431 (**b**) after X-ray exposure. Representative images (enlarged fragments of the tablet wells) of colonies NCTC L929 and A431 without exposure (0 Gy) and after exposure to X-ray radiation at doses of 3 and 6 Gy (**c**).

## Data Availability

The original contributions presented in the study are included in the article/Appendix A, further inquiries can be directed to the corresponding author.

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
