# Peer review of "Novel Flavin Mononucleotide-Functionalized Cerium Fluoride Nanoparticles for Selective Enhanced X-Ray-Induced Photodynamic Therapy"

_jfb, 2024, doi:10.3390/jfb15120373_

Round 1
Reviewer 1 Report
Comments and Suggestions for Authors
The authors present a novel application of flavin mononucleotide-functionalized cerium fluoride nanoparticles for X-ray-induced photodynamic therapy (X-PDT) in vitro. They have provided extensive data characterizing the physical and chemical properties of the nanoparticles. Furthermore, the study includes detailed in vitro investigations using both normal and cancer cell lines to evaluate the cytotoxicity of the nanoparticles and their efficacy in X-ray-induced PDT. These experiments contribute to a comprehensive understanding of the nanoparticles' biological interactions and therapeutic promise.
Suggestions for improvement:
- Clarification in line 40
The phrase, ‘In 2006 was suggested using X-ray radiation with the high tissue-penetrability to activate PS and called X-ray induced PDT (X-PDT)’, could be rephrased for clarity. - Figure 4 and the definition of IC value
A clearer explanation of how the IC20 value in Figure 4 was derived is essential. While "IC" typically refers to the inhibitory concentration, it is unclear why the bar was shown in the Y axis marking cell viability or live dead ratio. - Cytotoxicity discrepancies
The authors should explain the observed higher proportion of background dead cells in the A431 cancer cell line compared to the L929 normal cell line. - Differential cellular uptake
In the Discussion section, the authors briefly mention the differential cellular uptake of the nanoparticles between L929 and other cancer cells. It would be beneficial to confirm whether this phenomenon also applies to the A431 cell line. - Limitations and future directions
A dedicated paragraph discussing the limitations of the current study would greatly enhance its impact. For instance, the study could address the following points: - The challenges in translating in vitro findings to in vivo systems, including issues of biodistribution, off-target effects, and immune response.
- Potential limitations in the reproducibility of X-PDT efficacy across different tumor microenvironments or in varying physiological conditions.
Adding this discussion would provide a more comprehensive perspective and emphasize the translational relevance of the findings.
Comments on the Quality of English LanguageEnglish editing is required to improve the manuscript as some sentences do not make much sentence.
Author Response
Reviewer 1
Summary: Thank you very much for taking the time to review this manuscript. We greatly appreciate the comments you have provided, and we believe that your comments have helped us to substantially improve the quality of our manuscript. Please find the detailed responses below and the corresponding revisions highlighted in the re-submitted files. We have also provided the line numbers where the corresponding changes were made.
Comments 1: Clarification in line 40. The phrase, ‘In 2006 was suggested using X-ray radiation with the high tissue-penetrability to activate PS and called X-ray induced PDT (X-PDT)’, could be rephrased for clarity.
Response 1: A modification has been made to the aforementioned proposal in the Introduction section.
Comment 2: Figure 4 and the definition of IC value. A clearer explanation of how the IC20 value in Figure 4 was derived is essential. While "IC" typically refers to the inhibitory concentration, it is unclear why the bar was shown in the Y axis marking cell viability or live dead ratio.
Response 2: Thank you very much for your comment. We have removed the IC20 line for a clearer understanding and evaluation of the viability graphs (Figure 5).
Comments 3: Cytotoxicity discrepancies. The authors should explain the observed higher proportion of background dead cells in the A431 cancer cell line compared to the L929 normal cell line.
Response 3: We have added an explanation to the text starting at lines 482-499.
Comments 4: Differential cellular uptake. In the Discussion section, the authors briefly mention the differential cellular uptake of the nanoparticles between L929 and other cancer cells. It would be beneficial to confirm whether this phenomenon also applies to the A431 cell line.
Response 4: We have added an explanation to the text in the Discussion section (lines 520-521).
Comments 5: Limitations and future directions
A dedicated paragraph discussing the limitations of the current study would greatly enhance its impact. For instance, the study could address the following points:
The challenges in translating in vitro findings to in vivo systems, including issues of biodistribution, off-target effects, and immune response.
Potential limitations in the reproducibility of X-PDT efficacy across different tumor microenvironments or in varying physiological conditions.
Response 5: We have added some additional information to the Conclusion section (lines 582-586).
As the purpose of this paper was not an in vivo study, we did not discuss these issues in detail. Additionally, the editors have suggested that we should shorten the Discussion section. However, we are currently conducting in vivo studies using an allograft tumor mouse models to study the biodistribution of nanoparticles and other aspects. In our future work, we plan to explore these topics in greater depth.
Reviewer 2 Report
Comments and Suggestions for Authors
The manuscript by Popova, N. and co-workers describes synthesis of cerium fluoride nanoparticles modified with favin mononucleotide for X-PDT therapy and the study of their physicochemical properties. The authors have also investigated the cyto and genotoxicitiy, proliferative activity under the exposure of X-rays on normal and cancer cells.
In my opinion, the topic is very interesting, and the presented work is of interest to the researchers working in the field of materials chemistry related to cancer treatment and photodynamic therapy.
In my opinion, the presented work is within the standards that are needed for publication in Journal of Functional Biomaterials. However, before publication, authors should explain the discussion section more concisely. This section should be shortened. Lines 474-476 are repeated in 496-499. This needs to be corrected.
Line 502. Indicate the table number, TS?
Author Response
Reviewer 2
Summary: Thank you very much for taking the time to review this manuscript. We greatly appreciate the comments you have provided, and we believe that your comments have helped us to substantially improve the quality of our manuscript. Please find the detailed responses below and the corresponding revisions highlighted in the re-submitted files. We have also provided the line numbers where the corresponding changes were made.
Comments 1: However, before publication, authors should explain the discussion section more concisely. This section should be shortened.
Response 1: We have modified the Discussion section and reduced it by 29 lines (lines 414-560).
Comments 2: Lines 474-476 are repeated in 496-499. This needs to be corrected.
Response 2: Has been corrected in the Discussion section (lines 479-481).
Comments 3: Line 502. Indicate the table number, TS?
Response 3: Has been corrected (line 522).
Reviewer 3 Report
Comments and Suggestions for Authors
This work proposes a new type of nanomaterial based on CeF₃ and FMN as a photosensitizer and nanoscintillator for X-PDT, which is highly innovative. The study combines physical and chemical characterization (such as TEM, DLS, XRD) and biological experiments (MTT, clone formation experiments, etc.) to systematically evaluate the physical properties, biocompatibility and anti-tumor effects of nanomaterials. The experimental results of ROS generation, cytotoxicity and genotoxicity are discussed in detail, and the data are clear. However, there are still some issues that need improvement:
1. The article mentions that "scintillation nanomaterials exhibits good biocompatibility and efficiently absorbs ionizing radiation", and it is recommended to provide more literature support.
2. In Figure 3, the mechanism of ROS increase under low pH conditions is not fully discussed. It is recommended to supplement the discussion on the relationship between the change of Ce³⁺/Ce⁴⁺ ratio and the pH of the microenvironment and the specific mechanism of its selective toxicity to cancer cells.
3. The results of the MTT experiment showed that normal cells showed a transient decrease in activity under certain conditions, but the possible reasons or potential mechanisms were not analyzed in depth. It is recommended to add a discussion on this phenomenon.
4. The article mentions the potential of CeF₃-FMN in X-PDT, but the challenges it may face in actual tumor treatment (such as distribution in the body, targeting, and long-term safety) are not discussed enough. It is recommended to add a prospective analysis of its potential for translational medicine.
Author Response
Reviewer 3
Summary: Thank you very much for taking the time to review this manuscript. We greatly appreciate the comments you have provided, and we believe that your comments have helped us to substantially improve the quality of our manuscript. Please find the detailed responses below and the corresponding revisions highlighted in the re-submitted files. We have also provided the line numbers where the corresponding changes were made.
Comments 1: The article mentions that "scintillation nanomaterials exhibits good biocompatibility and efficiently absorbs ionizing radiation", and it is recommended to provide more literature support.
Response 1: We have added additional links in the Introduction section (line 60).
Comments 2: In Figure 3, the mechanism of ROS increase under low pH conditions is not fully discussed. It is recommended to supplement the discussion on the relationship between the change of Ce³⁺/Ce⁴⁺ ratio and the pH of the microenvironment and the specific mechanism of its selective toxicity to cancer cells.
Response 2: We have added an explanation to the text (lines 495-499).
Comments 3: The results of the MTT experiment showed that normal cells showed a transient decrease in activity under certain conditions, but the possible reasons or potential mechanisms were not analyzed in depth. It is recommended to add a discussion on this phenomenon.
Response 3: Indeed, we observed a slight decrease in cell viability after 48 hours of co-incubation. However, after 24 and 72 hours, no such effects were observed. Given the low expression of riboflavin receptors in mouse fibroblasts of the L929 line, it is assumed that the endocytosis process of these nanoparticles takes more than 12-16 hours. This is due to the composition and size of the nanoparticles, as well as the average doubling time of the cells, which is 21 hours (Torabi et al., 2023 doi: 10.22092/ARI.2023.361584.2663). Thus, we expect the first doubling of the cell culture to occur between 24 and 48 hours after the endocytosis process. Thus, we observed a slight decrease in the MTT index of the test, less than 10%, at 48 hours after co-incubation. At this point, the standard doubling time had shifted. It is worth noting that the MTT method is based on the analysis of the activity levels of cytoplasmic and mitochondrial dehydrogenases. The optical density of formazan directly correlates with the number of cells in the well. Finally, we can conclude that the absence of negative effects of nanoparticles after 72 hours confirms the high biocompatibility of these nanoparticles for normal L929 cells. The decrease observed at 48 hours was not critical for the viability of the culture.
We have added a short explanation to the text (lines 486-496).
Comments 4: The article mentions the potential of CeF₃-FMN in X-PDT, but the challenges it may face in actual tumor treatment (such as distribution in the body, targeting, and long-term safety) are not discussed enough. It is recommended to add a prospective analysis of its potential for translational medicine.
Response 4: We have added some additional information to the Conclusion section (lines 584-587).
Reviewer 4 Report
Comments and Suggestions for Authors
While the size of the nanoparticles is provided, the concentration units (10⁻⁷ and 10⁻³ M) could be supplemented with mass or molar equivalence for easier comparison.
The "significant reduction in clonogenic activity" could benefit from specific percentages or quantitative data to underscore the therapeutic efficacy.
The manuscript mentions "pH-dependent radiation-induced redox activity" but does not elaborate on the pH range or its physiological relevance.
Further clarity on how FMN contributes to the selective toxicity and radioprotective effects would be valuable.
While it is stated that the NPs are non-toxic to normal cells but toxic to tumor cells, specific cytotoxicity percentages or IC50 values would enhance the claim.
The potential for clinical translation, scalability of the synthesis method, and cost-effectiveness of the technology are not addressed.
How do CeF₃-FMN NPs compare to existing radiosensitizers or X-PDT agents in terms of efficiency, selectivity, and safety? A brief mention could strengthen the novelty claim.
The statement that the NPs act as radioprotectors for normal cells requires clarification or supporting evidence since this dual functionality might raise questions about the mechanism.
The Manuscript is scientifically sound and highlights a promising development in X-PDT. Addressing the minor issues listed above would enhance its impact and ensure clarity for the reader.
Author Response
Reviewer 4
Summary: Thank you very much for taking the time to review this manuscript. We greatly appreciate the comments you have provided, and we believe that your comments have helped us to substantially improve the quality of our manuscript. Please find the detailed responses below and the corresponding revisions highlighted in the re-submitted files. We have also provided the line numbers where the corresponding changes were made.
Comments 1: While the size of the nanoparticles is provided, the concentration units (10⁻⁷ and 10⁻³ M) could be supplemented with mass or molar equivalence for easier comparison.
Response 1: Information about the concentration units is presented in Materials and Methods section (2.1. Synthesis of CeF3 and CeF3-FMN nanoparticles) (lines 95-96).
«Nanoparticles sol stock concentration is 45,2 × 10-3 М = 10,1 mg/mL for CeF3, 42,55 × 10-3 М = 8,2 mg/mL CeF3-FMN».
Comments 2: The "significant reduction in clonogenic activity" could benefit from specific percentages or quantitative data to underscore the therapeutic efficacy.
Response 2: Has been corrected in the Abstract section (lines 26-27). Furthermore, the clonogenic activity of cells was calculated using the 37% survival dose (D0), which is presented in the Supplementary file, Table S3.
Comments 3: The manuscript mentions "pH-dependent radiation-induced redox activity" but does not elaborate on the pH range or its physiological relevance.
Response 3: We have added the necessary information to the Discussion section (lines 489-495).
Comments 4: Further clarity on how FMN contributes to the selective toxicity and radioprotective effects would be valuable.
Response 4: We have added the necessary information to the Introduction section (lines 72-75).
Comments 5: While it is stated that the NPs are non-toxic to normal cells but toxic to tumor cells, specific cytotoxicity percentages or IC50 values would enhance the claim.
Response 5: We would like to point out that, without irradiation, nanoparticles do not cause a significant cytotoxic effect on both types of nanoparticles. The IC50 values were not determined. However, we have observed a trend that suggests cancer cells may be more sensitive to the effects of nanoparticles after irradiation. Using clonogenic analysis, we have detected a significant effect.
Comments 6: The potential for clinical translation, scalability of the synthesis method, and cost-effectiveness of the technology are not addressed.
Response 6: The principal objective of this study was to develop and validate a method for the synthesis of novel cerium fluoride nanoparticles that have been modified with FMN. At this juncture in the study, a scalable method for nanoparticle synthesis has yet to be developed, as this was not a primary objective at this stage. It is, however, noteworthy that the proposed technique for nanoparticle synthesis allows for the production of high concentrations of these nanoparticles.
Comments 7: How do CeF₃-FMN NPs compare to existing radiosensitizers or X-PDT agents in terms of efficiency, selectivity, and safety? A brief mention could strengthen the novelty claim.
Response 7: We have added additional information about existing X-PDT agents in the Introduction section (lines 50-56).
Comments 8: The statement that the NPs act as radioprotectors for normal cells requires clarification or supporting evidence since this dual functionality might raise questions about the mechanism.
Response 8: The definition of a radioprotector has been updated to an antioxidant in the Abstract section (line 28).